

# Bioinformatics analysis of oxidative stress genes in the pathogenesis of ulcerative colitis based on a competing endogenous RNA regulatory network

Qifang Li[1], Yuan Liu[2], Bingbing Li[2], Canlei Zheng[2], Bin Yu[2], Kai Niu[2] and Yi Qiao[3]

[1] Department of Traditional Chinese Medicine, Affiliated Hospital of Jining Medical University, Jining, Shandong, China
[2] College of Integrated Chinese and Western Medicine, Jining Medical University, Jining, Shandong, China
[3] School of Public Health, Jining Medical University, Jining, Shandong, China

## ABSTRACT

**Background:** Ulcerative colitis (UC) is a common chronic disease associated with inflammation and oxidative stress. This study aimed to construct a long noncoding RNA (lncRNA)-microRNA (miRNA)-messenger RNA (mRNA) network based on bioinformatics analysis and to explore oxidative stress-related genes underlying the pathogenesis of UC.

**Methods:** The GSE75214, GSE48959, and GSE114603 datasets were downloaded from the Gene Expression Omnibus database. Following differentially expressed (DE) analysis, the regulatory relationships among these DERNAs were identified through miRDB, miRTarBase, and TargetScan; then, the lncRNA-miRNA-mRNA network was established. The Molecular Signatures Database (MSigDB) was used to search oxidative stress-related genes. Gene Ontology (GO) and Kyoto Encyclopedia of Genes and Genomes (KEGG) analyses were performed for functional annotation and enrichment analyses. Based on the drug gene interaction database DGIdb, drugs that interact with oxidative stress-associated genes were explored. A dextran sulfate sodium (DSS)-induced UC mouse model was used for experimental validation.

**Results:** A total of 30 DE-lncRNAs, 3 DE-miRNAs, and 19 DE-mRNAs were used to construct a lncRNA-miRNA-mRNA network. By comparing these 19 DE-mRNAs with oxidative stress-related genes in MSigDB, three oxidative stress-related genes (*CAV1, SLC7A11*, and *SLC7A5*) were found in the 19 DEM sets, which were all negatively associated with miR-194. GO and KEGG analyses showed that *CAV1, SLC7A11*, and *SLC7A5* were associated with immune inflammation and steroid hormone synthesis. In animal experiments, the results showed that dexamethasone, a well-known glucocorticoid drug, could significantly decrease the expression of *CAV1, SLC7A11*, and *SLC7A5* as well as improve UC histology, restore antioxidant activities, inhibit inflammation, and decrease myeloperoxidase activity.

**Conclusion:** *SLC7A5* was identified as a representative gene associated with glucocorticoid therapy resistance and thus may be a new therapeutic target for the treatment of UC in the clinic.

Corresponding author
Bin Yu, yubinsd@yeah.net

## INTRODUCTION

Inflammatory bowel disease, which is a kind of autoimmune disease, mainly includes two subtypes: ulcerative colitis (UC) and Crohn's disease (CD) (*Ng et al., 2017*). In CD, all layers of the bowel wall are inflamed; in contrast, UC is generally characterized by mucosal layer inflammation and damage to the superficial bowel wall (*Hibi & Ogata, 2006*; *Kobayashi et al., 2020*). The disease course of UC usually involves remission and exacerbation in alternating cycles, and if treatment is not performed in a timely, colorectal cancer can develop (*Ungaro et al., 2017*; *Lissner & Siegmund, 2013*). From the year 1955, glucocorticoids have been considered effective for patients with UC, and the use of glucocorticoids dramatically decreases the mortality of patients with moderate-to-severe UC (*Nakase et al., 2021*). However, long-term use of corticosteroids not only induces glucocorticoid-resistance but also leads to numerous adverse effects such as depression, cataracts, and osteoporosis (*Nakase et al., 2021*; *Magro et al., 2017*; *Truelove & Witts, 1955*). Therefore, elucidating the pathogenesis of the initiation and progression of UC is imperative to develop novel therapies.

The development of UC involves multiple mechanisms, and oxidative stress caused by the imbalance between antioxidants and oxidants plays a vital role (*Grisham, 1994*). The presence of excessive reactive oxygen species (ROS) including peroxynitrite, hydrogen peroxide, and superoxide can reduce the productions of endogenous antioxidants, eventually resulting in cell death *via* oxidative damage to DNA, membrane lipids, and cellular proteins (*Amirshahrokhi, Bohlooli & Chinifroush, 2011*; *Ferrat et al., 2019*; *Niu et al., 2015*). The reaction of DNA with ROS leads to the modification of DNA bases and contributes to subsequent carcinogenesis. For example, ROS may interact with genomic DNA to generate some base modifications with pro-mutagenic potentials such as 8-nitro-2′-deoxyguanosine (8-NO$_2$-dG) and 8-oxo-7,8-dihydro-2′-deoxyguanosine (8-oxodG) (*Kaneko et al., 2008*; *Kondo et al., 1999*). One previous study demonstrated excessive production of 8-oxodG in patients with UC-associated carcinogenesis (*Gushima et al., 2009*). This series of genetic changes serves as a trigger in the pathogenesis of chronic inflammation-associated diseases. Other studies have demonstrated local DNA damage in the colon and systemic DNA damage involving the hepatocytes, lymphoid organs, and blood in mice with dextran sulfate sodium (DSS)-induced UC, and these outcomes are considered to be partly mediated by oxidative stress (*Westbrook et al., 2011*, *2009*; *Trivedi & Jena, 2012*). In addition, the expression of some inflammatory genes, such as TNF-α, is reported to be regulated by oxidative stress-associated genes, and the application of TNF-α inhibitors has been proven to be effective for the treatment of UC (*Verhasselt, Goldman & Willems, 1998*; *Barrie & Regueiro, 2007*). Therefore, identifying oxidative stress-associated genes and establishing a direct linkage between these genes and UC pathogenesis may be helpful for the treatment of patients with UC.

Evidence is now emerging to indicate that non-coding RNAs play vital roles in inflammatory bowel disease, especially in the progression of UC (*Ghafouri-Fard, Eghtedarian & Taheri, 2020*; *Schaefer, 2016*). Non-coding RNAs are RNAs that are unable to code proteins and can regulate multiple biological processes through modulating the expression of coding RNAs (*Quinn & Chang, 2016*; *Schmitz, Grote & Herrmann, 2016*; *Matsui & Corey, 2017*). There are mainly four types of non-coding RNAs: long non-coding RNAs (lncRNAs), microRNAs (miRNAs), circular RNAs (circRNAs), and extracellular RNAs (exRNAs) (*St Laurent, Wahlestedt & Kapranov, 2015*; *Ling, Fabbri & Calin, 2013*; *Sato-Kuwabara et al., 2015*; *Ebbesen, Kjems & Hansen, 2016*). *Salmena et al. (2011)* proposed the notion of a competing endogenous RNA (ceRNA) regulatory network that involves interactions among these RNAs. lncRNAs are RNA molecules that can affect the transcriptional and post-transcriptional expression of genes (*Quinn & Chang, 2016*; *Schmitz, Grote & Herrmann, 2016*). Increasing evidence has demonstrated that lncRNAs have several molecular functions, such as being involved in regulatory transcription and acting as miRNA sponges and regulatory RNA binding proteins (*Jain et al., 2017*; *Zacharopoulou et al., 2017*; *Khan et al., 2022*). In general, lncRNAs can sponge miRNA through miRNA response elements, which eventually affect the binding between miRNAs and messenger RNAs (mRNAs) (*Zhang et al., 2019*; *Ala, 2020*). lncRNAs have also been reported to be involved in regulating several kinds of human diseases including neurological diseases, cancers, cardiovascular diseases, and inflammatory bowel disease (*Ghafouri-Fard, Eghtedarian & Taheri, 2020*; *Canseco-Rodriguez et al., 2022*; *Chi et al., 2019*; *Poller et al., 2018*). Furthermore, a lncRNA-related ceRNA network has been identified as a key mechanism involved in immune-related diseases including systemic lupus erythematosus (*Song et al., 2021*) and rheumatoid arthritis (*Zhang et al., 2020*). In addition, *Dong et al. (2022)* constructed a lncRNA-related ceRNA network in UC and identified two mRNAs (CTLA1 and STAT1) that are associated with immune cell infiltration. However, research on oxidative stress genes underlying the lncRNA-related ceRNA network in UC is still in the preliminary stages.

In the current study, the original data from the UC group and control group were obtained from the NCBI Gene Expression Omnibus (GEO) database. Based on an analysis of differential gene expression, differentially expressed (DE)-lncRNAs, DE-miRNAs, and DE-mRNAs were identified. This study aimed to probe a complete lncRNA-miRNA-mRNA network to determine the roles of oxidative stress genes in the pathogenesis and drug resistance mechanism of UC. Subsequently, a DSS-induced mouse model was established to validate the results of the bioinformatics analysis. These findings clarified the relationships among the identified oxidative stress genes and the pathogenesis and drug resistance mechanism in UC, providing some references for the clinical diagnosis and treatment of UC.

## MATERIALS AND METHODS

### Data resource

A total of three datasets were downloaded from the GEO database (http://www.ncbi.nlm.nih.gov/geo). In detail, the lncRNA/mRNA expression profile is
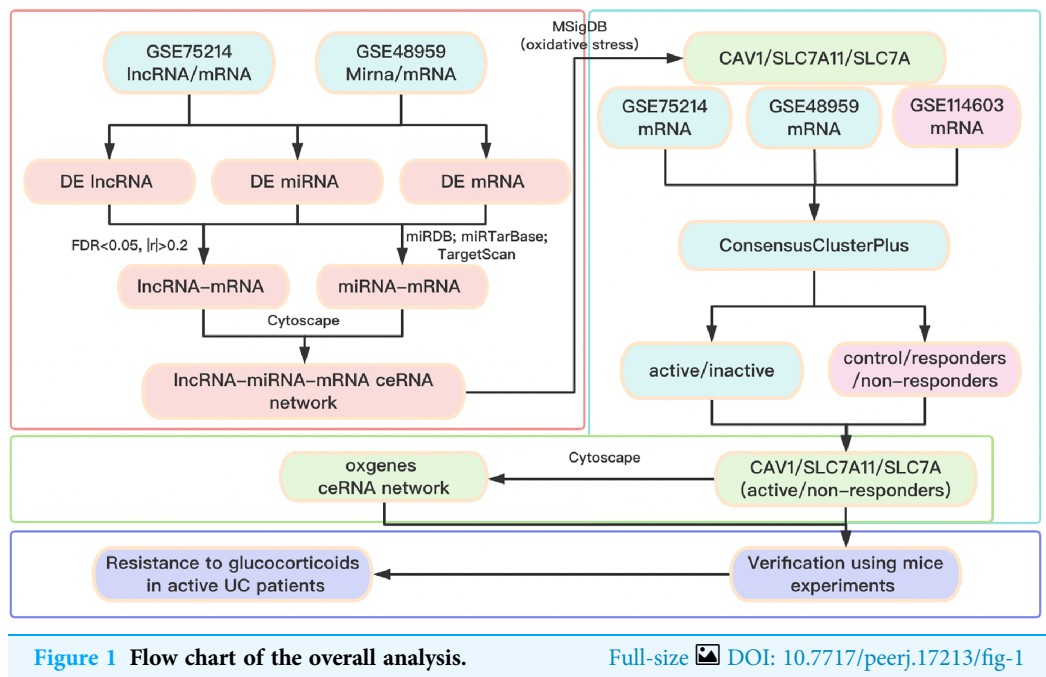

**Figure 1 Flow chart of the overall analysis.**

GSE75214 (https://doi.org/10.6084/m9.figshare.25263436.v1. A-B), the miRNA/mRNA expression profile is GSE48959 (https://doi.org/10.6084/m9.figshare.25263436.v1. C-D), and the mRNA expression profile of glucocorticoid-resistant genes is GSE114603 (https://doi.org/10.6084/m9.figshare.25263436.v1. E). The clinical information of patients with UC from the GEO database is shown in Table S1. The software package is available on GitHub at https://github.com/Yubinnet/UC. A flowchart of the bioinformatics analysis is depicted in Fig. 1, among which GSE75214 dataset includes the lncRNA/mRNA expression profile, and GSE48959 dataset includes the miRNA/mRNA expression profile. Patient consent was not required for this work because all the datasets originated from a free open-access database on the internet.

## Identification of DE-lncRNAs, DE-miRNAs, and DE-mRNAs

The factoextra package of R (https://github.com/kassambara/factoextra) was utilized to perform a principal component analysis (PCA) of gene expression using the fviz_pca_ind function. We then integrated the analysis of lncRNA/mRNA and miRNA/mRNA and used the limma package of R to screen DE-lncRNAs, DE-miRNAs, and DE-mRNAs between UC and control samples. We set lncRNA ($P < 0.01$, |log2(fold change)| > 0.58), miRNA ($P < 0.05$, |log2(fold change)| > 0.58), and mRNA ($P < 0.05$, |log2(fold change)| > 0.58) as the cutoff point to select DE-lncRNAs, DE-miRNAs, and DE-mRNAs, respectively.

## Construction of ceRNA network

First, we used the cor.test function of R to calculate the Spearman correlation between DE-lncRNAs and DE-mRNAs according to their expression levels. The Benjamin & Hochberg method was used to calculate the false discovery rate (FDR). An FDR < 0.05 plus | r | > 0.2 was used to determine significant association pairs of lncRNA-mRNA. Based on

miRDB (http://mirdb.org/index.html), miRTarBase (https://mirtarbase.cuhk.edu.cn/~miRTarBase/miRTarBase_2022/php/index.php), and TargetScan (https://www.targetscan.org/vert_80/), each DEmiRNA's predicted mRNA was obtained. Then, based on the expression status of DEmiRNAs and DEmRNAs between UC and control samples, miRNA-mRNA association pairs were selected for subsequent analysis. Finally, the intersections of lncRNA-mRNA association pairs with miRNA-mRNA association pairs were obtained to construct the lncRNA-miRNA-mRNA ceRNA network, which was visualized using Cytoscape version 3.8.1 (https://cytoscape.org/) (*Shannon et al., 2003*).

## Identification of oxidative stress-associated genes

We searched the Molecular Signatures Database (MSigDB, v7.4) (http://software.broadinstitute.org/gsea/msigdb) using the search term "oxidative stress" and selected all the obtained genes as oxidative stress-related gene sets (https://doi.org/10.6084/m9.figshare.25263436.v1. F). Considering the relationship between oxidative stress-associated genes and inflammatory processes, we further explored the relationship between oxidative stress-associated genes and the pathogenesis of UC. Based on the R package ConsensusClusterPlus (http://bioconductor.org/packages/release/bioc/html/ConsensusClusterPlus.html), we conducted a consistency cluster analysis for UC samples, with the following parameters: maxK = 10 (maximum cluster number to evaluate) and reps = 100 (number of samples). An appropriate number of clusters was selected to verify the classification reliability of different subtypes of UC (active UC and inactive UC). Then, we conducted enrichment analysis on whether UC patients are in active phase and UC subtypes (fisher. test). To acquire representative genes of UC subtypes, we used the limma package of R to analyze the differences between a particular subtype sample and other subtype samples. Only genes that were highly expressed in this subtype (log2(fold change) > 1, $P < 0.05$) were selected as representative genes of this subtype. Then, we used cor.test to calculate the Spearman correlation coefficient between the representative genes of each subtype and oxidative stress-associated genes in UC samples. Based on the parameters of an FDR < 0.05 and $|R| > 0.8$, the representative genes that highly interacted with oxidative stress-associated genes in each subtype were selected. Finally, Gene Ontology (GO) and Kyoto Encyclopedia of Genes and Genomes (KEGG) analyses were performed for functional annotation and enrichment analyses. A $P$ value < 0.05 was regarded as a statistically significant difference.

## Prediction of drug interactions

Based on the drug gene interaction database DGIdb (https://www.dgidb.org/search_interactions), drugs that interact with oxidative stress-associated genes were further explored.

## Animals and treatment

This study was performed in accordance with The ARRIVE Guidelines 2.0: Updated Guidelines for Reporting Animal Research. A total of 28 6-week-old C57BL/6J mice weighing 18–20 g were obtained from Pengyue Laboratory Animal Breeding Co., Ltd.

(Jinan, China). The mice were maintained in a specific pathogen-free animal laboratory and housed using standard cages in a room with a humidity of 50 ± 20%, a temperature of 23 ± 3 °C, and a 12 h light/12 h dark cycle. All animals had free access to standard laboratory food and water. They were separated into four groups at random with seven mice in each group: control, DSS, DSS + normal saline (NS), and DSS + dexamethasone (DXM) groups. The control group received 50 μL of NS *via* intraperitoneal injection for 7 days. The DSS group received 3.5% weight/volume of DSS *via* intraperitoneal injection (50 μL; MP Biomedicals, Santa Ana, CA, USA) for 7 days. The DSS + NS group received an intraperitoneal injection of DSS (50 μL) for 7 days and then 50 μL of NS for 7 days. The DSS + DXM group received an intraperitoneal injection of DSS (50 μL) for 7 days and then 1.2 mg/kg of DXM (Sigma, St. Louis, MO, USA) for 7 days. All experimental procedures were conducted in compliance with the Guidelines for Care and Use of Laboratory Animals of the National Institutes of Health and approved by the Ethics Committee of Jining Medical University (approval number: JNMC-2023-DW-090).

## Histological examination

After administration, mice in the different groups were anesthetized with an intraperitoneal injection of 5% pentobarbital (0.2 mL/10 g) and sacrificed by cervical dislocation. The colonic tissues were collected for hematoxylin & eosin (HE) staining *via* a commercial kit (Beyotime, Shanghai, China). The colon samples were fixed, dehydrated, and embedded in paraffin. After cutting into 4-μm-thick slices, dewaxing, and rehydration, the slices were prepared for HE staining. A histological examination of the colonic tissues was performed using a light microscope (OLYMPUS, Tokyo, Japan), with a magnification of ×100.

## Total RNA isolation and qRT-PCR

qRT-PCR was performed in accordance with The Minimum Information for Publication of Quantitative Real-Time PCR Experiments (MIQE) guidelines. A Total RNA Extraction Kit (Promega, Madison, WI, USA) was used to isolate the total RNA from the colonic tissue samples (50 mg) from the control, DSS, DSS + NS, and DSS + DXM groups ($n = 7$). RNA purity was measured using NanoDrop (Peqlab Biotechnologie GmbH, Erlangen, Germany). The OD260/280 ratio was used as an indicator for RNA purity. A ratio higher than 1.8 was regarded as suitable for gene expression measurements. The GoScript reverse transcription system (Promega Corporation, Madison, WI, USA) was used to reverse transcribe the extracted RNA (1 μg) into cDNA at 42 °C for 45 min. With the aid of Hifair® II 1st Strand cDNA Synthesis SuperMix (Yeasen Biotechnology, Shanghai, China) and Hieff® qPCR SYBR Green Master Mix (Yeasen Biotechnology, Shanghai, China), qRT-PCR analysis was performed on an ABI 7900 Real-Time PCR System (Applied Biosystems, Foster City, CA, USA). The thermocycling conditions were as follows: initial denaturation for 10 min at 95 °C; 40 cycles of 95 °C for 15 s and 60 °C for 30 s; and final extension for 1 min at 60 °C. Gene expression levels were calculated with the $2^{-\Delta\Delta Ct}$ approach with GAPDH or U6 for normalization. We designed the primers using

**Table 1 Real-time PCR primer synthesis list.**

| Gene | Sequences | |
| --- | --- | --- |
| COX-2 | Forward | 5′-AAGACTACGTGCAACACCTGAG-3′ |
| | Reverse | 5′-GTGCCAGTGATAGAGTGTGT-3′ |
| SOCS3 | Forward | 5′-GGACCAAGAACCTACGCATCCA-3′ |
| | Reverse | 5′-CACCAGCTTGAGTACACAGTCG-3′ |
| IL-6 | Forward | 5′-GACAAAGCCAGAGTCCTTCAGAGA-3′ |
| | Reverse | 5′-CTAGGTTTGCCGAGTAGATCTC-3′ |
| TLR4 | Forward | 5′-TCCACTGGTTGCAGAAAATGC-3′ |
| | Reverse | 5′-TCATCAGGGACTTTGCTGAGTTT-3′ |
| TNF-α | Forward | 5′-CCCTCACACTCACAAACCAC-3′ |
| | Reverse | 5′-ACAAGGTACAACCCATCGGC-3′ |
| IL-1β | Forward | 5′-TGGACCTTCCAGGATGAGGACA-3′ |
| | Reverse | 5′-GTTCATCTCGGAGCCTGTAGTG-3′ |
| Foxp3 | Forward | 5′-TTCGCCTACTTCAGAAACCACC-3′ |
| | Reverse | 5′-ATTCATCTACGGTCCACACTGCT-3′ |
| Cav1 | Forward | 5′-ACGTAGACTCCGAGGGACATC-3′ |
| | Reverse | 5′-CGTCGTCGTTGAGATGCTTG-3′ |
| Slc7a5 | Forward | 5′-ATGGAGTGTGGCATTGGCTT-3′ |
| | Reverse | 5′-GAGCACCGTCACAGAGAAGAT-3′ |
| IL-17 | Forward | 5′-TGACCCCTAAGAAACCCCCA-3′ |
| | Reverse | 5′-TCATTGTGGAGGGCAGACAA-3′ |
| Slc7a11 | Forward | 5′-TCCTCTGACGATGGTGATGC-3′ |
| | Reverse | 5′-GCTGAATGGGTCCGAGTAAAG-3′ |
| MPO | Forward | 5′-CGTGTCAAGTGGCTGTGCCTAT-3′ |
| | Reverse | 5′-AACCAGCGTACAAAGGCACGGT-3′ |
| GAPDH | Forward | 5′-GGCATGGACTGTGGTCATGAG-3′ |
| | Reverse | 5′-TGCACCACCAACTGCTTAGC-3′ |

Primer-BLAST (http://www.ncbi.nlm.nih.gov/tools/primer-blast). MFEprimer-2.0 (https://www.mfeprimer.com/old-versions/mfeprimer-2.0/) was used to perform specificity checking. The primers utilized in this work are shown in Table 1.

## Myeloperoxidase (MPO) activity

MPO activity was measured using a sandwich enzyme immunoassay commercial kit (BIOXYTECH® MPO-EIATM, OXIS Health Products; Inc Oxis-International, Portland, OR, USA), according to the manufacturer's instructions.

## Detection of the levels of GSH, GSSG, ROS, SOD, and HMGB-1

According to the instructions of the Reduced Glutathione (GSH) Content Assay Kit (Sangon Biotech, Shanghai, China), Oxidized Glutathione (GSSG) Assay Kit (Sangon Biotech, Shanghai, China), Reactive Oxygen Species (ROS) Assay Kit (Beyotime,

Beijing, China), Superoxide Dismutase (SOD) Activity Assay Kit (Sangon Biotech, Shanghai, China), and Mouse HMGB-1 ELISA Kit (Sangon Biotech, Shanghai, China), the levels of GSH, GSSG, SOD, and HMGB-1, respectively, were determined.

## Flow cytometry analysis of ROS

The level of ROS was assessed using flow cytometry analysis, employing a ROS Detection Kit based on a DCFH-DA probe (Beyotime, Beijing, China). Fresh colonic tissues were immediately placed in precooled PBS to clean them from blood and other pollutants. The tissues were dissected into small fragments of approximately 1 mm$^3$ using ophthalmic scissors, and they were subsequently immersed in precooled PBS and thoroughly rinsed to eliminate any residual cellular debris. The appropriate volume of enzyme digestion solution was added, and the cells were digested at 37 °C for 30 min. Intermittent agitation was performed during the digestion process. The digestion was terminated using PBS, followed by removal of the tissue mass through filtration with a 300-mesh nylon mesh. The filtered cells were collected and centrifuged at 500 g for 10 min. The supernatant was decanted, and the precipitate was washed twice with PBS. A solution of DCFH-DA at a final concentration of 10 µM was prepared by diluting it with FBS-free medium. After harvesting and washing the cells twice with PBS, they ($1 \times 10^6$) were incubated with 500 µL of 10 µM DCFH-DA for 20 min at 37 °C. The cells were subsequently washed three times with FBS-free medium. Finally, detection and analysis of cellular ROS levels were performed using a FACScan flow cytometer (BD Biosciences, Franklin Lakes, NJ, USA) equipped with an excitation wavelength of 488 nm.

## Western blotting

Total proteins were extracted from colonic tissues using RIPA lysis buffer (Beyotime, Beijing, China), and their concentrations were measured with a BCA Kit (Beyotime, Beijing, China). The collected proteins were then separated by 10% SDS polyacrylamide gel electrophoresis and transferred onto a PVDF membrane. The membrane was blocked with nonfat milk (5%), on which the primary antibodies NF-κB (1:1,000; Abcam, Cambridge, UK), IκB (1:1,000; Abcam, Cambridge, UK), IL-6 (1:1,000; Abcam, Cambridge, UK), TNF-α (1:1,000; Abcam, Cambridge, UK), and GAPDH (1:1,000; Abcam, Cambridge, UK), and the corresponding secondary antibody (1:3,000; Abcam, Cambridge, UK) were incubated. The signals were identified using an ECL Kit (Beyotime, Beijing, China), and immunoblots were quantified with Alpha Innotech software (Alpha Innotech, San Leandro, CA, USA). GAPDH was the normalization for proteins.

## Statistical analysis

The variations among the data were assessed by applying one-way ANOVA followed by Tukey's multiple comparison test. Data analysis was performed in SPSS software v22.0. The data are reported as the mean ± standard deviation (SD). $P < 0.05$ was considered statistically significant.

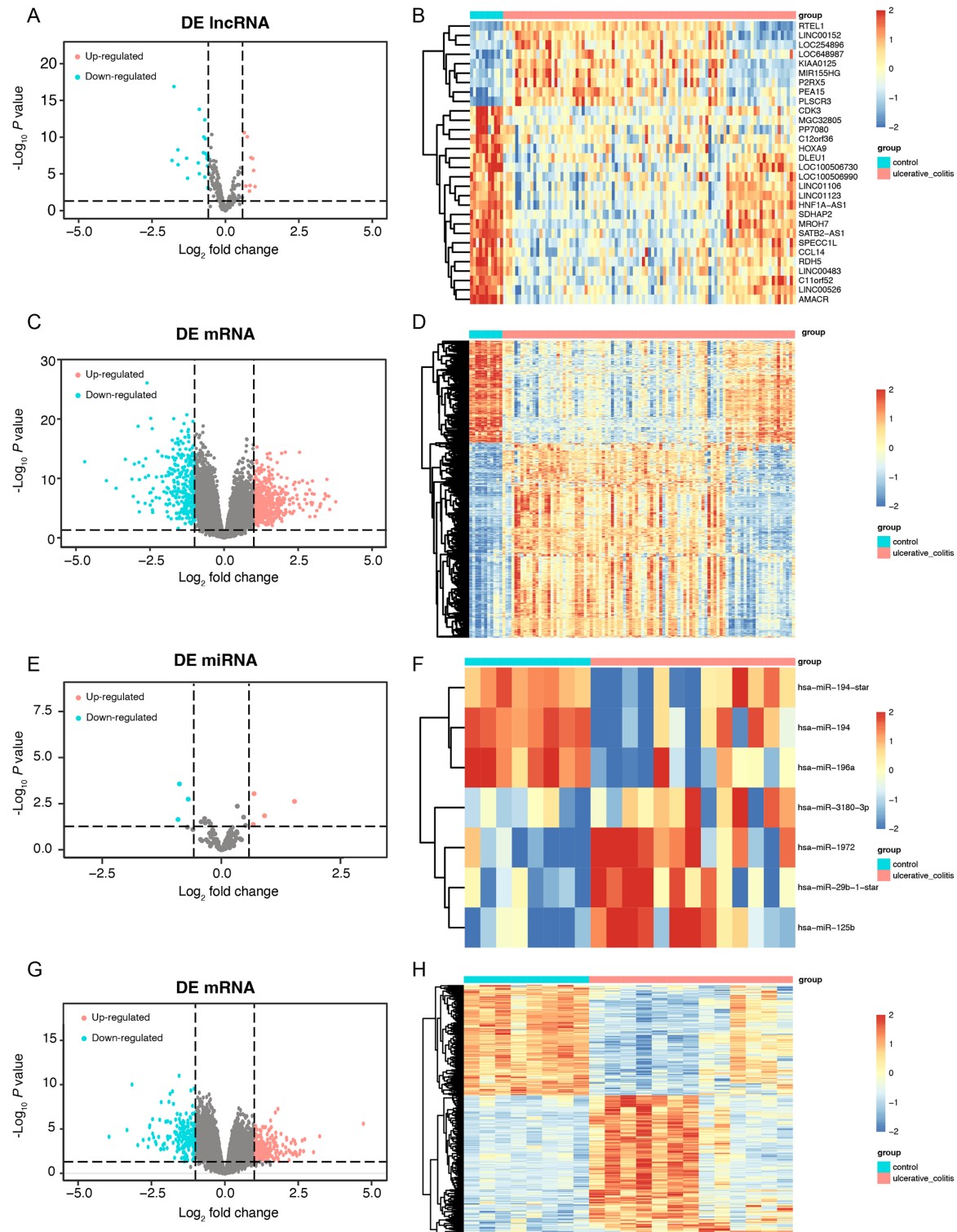

**Figure 2 The data integration and analysis for lncRNA/mRNA and miRNA/mRNA association pairs.** Volcano plots (A) and heat map (B) of DE-lncRNAs in the GSE75214 dataset. Volcano plots (C) and heat map (D) of DE-mRNAs in the GSE75214 dataset. Volcano plots (E) and heat map (F) of DE-miRNAs in the GSE48959 dataset. Volcano plots (G) and heat map (H) of DE-mRNAs in the GSE48959 dataset.

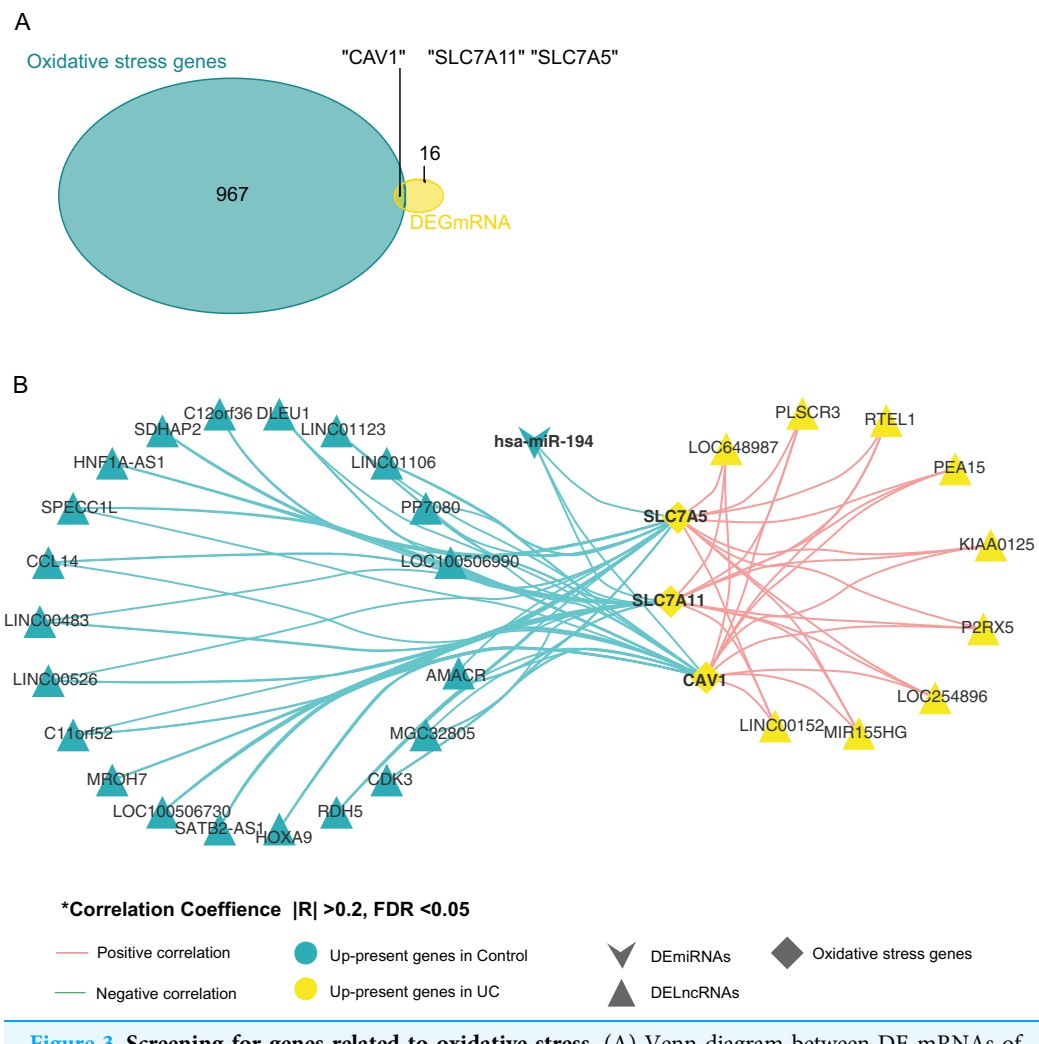

**Figure 3 Screening for genes related to oxidative stress.** (A) Venn diagram between DE-mRNAs of ceRNAs regulatory network and oxidative stress-related genes in MSigDB. (B) CeRNAs regulatory network based on oxidative stress-related genes.

## RESULTS

### Data integration and analysis for lncRNA/mRNA and miRNA/mRNA association pairs

First, we conducted a PCA for the lncRNA/mRNA data to elucidate expression differences between control and UC samples and found that there were significant differences (Figs. S1A and S1B, Tables S2A and S2B). The R packet "limma" was used to perform a differential analysis for the GSE75214 dataset. Based on the cutoff point of a *P* value < 0.05 plus |log2(fold change)| > 0.58, a total of 30 DE-lncRNAs were obtained, of which 9 lncRNAs were highly expressed, and 21 lncRNAs were slightly expressed in the UC samples (Figs. 2A and 2B, Table S3A). In addition, a total of 2,405 DEmRNAs were obtained. Among them, 1,394 mRNAs were confirmed to be upregulated, while 1,011 mRNAs were downregulated in UC samples (Figs. 2C and 2D, Table S3B). A PCA of the miRNA/mRNA data was subsequently performed (Figs. S1C and

S1D, Tables S2C and S2D). We found that there were significant differences in miRNA/mRNA between the control and UC samples. In the GSE48959 dataset, with the aid of the R packet "limma" ($P < 0.01$ & |log2(fold change)| > 0.58), a total of 7 DEmiRNAs including four upregulated and three downregulated miRNAs were identified in the UC samples (Figs. 2E and 2F, Table S3C). Moreover, 1,841 DEmRNAs were identified. Among these DEmRNAs, a total of 1,007 mRNAs were observed to be upregulated, while 834 were downregulated in the UC samples (Figs. 2G and 2H, Table S3D).

## Establishment of a ceRNA regulatory network

To better understand the molecular mechanism of DE-lncRNAs involved in the process of UC, nine upregulated lncRNAs and 21 downregulated lncRNAs were used to construct a lncRNA-miRNA-mRNA regulatory network. As shown in Table S4, this ceRNA network includes nine upregulated lncRNAs, 21 downregulated lncRNAs, two upregulated miRNAs, one downregulated miRNA, 14 upregulated mRNAs, and five downregulated mRNAs. We then constructed a random forest classifier based on the expression of these 19 DE-mRNAs in the GSE75214 and GSE48959 datasets, with 70% of the samples as the training set and 30% of the samples as the verification set. As illustrated in Fig. S2, both datasets had very high AUC density curves, suggesting that these mRNAs play key roles in the pathogenesis of UC.

## Screening for genes related to oxidative stress

By comparing oxidative stress genes (https://doi.org/10.6084/m9.figshare.25263436.v1 F) with these 19 DE-mRNAs, we found that there were three oxidative stress-related genes in the 19 DE-mRNA sets, namely *CAV1, SLC7A11*, and *SLC7A5* (Fig. 3A). These three mRNAs involved in the ceRNA network are shown in Fig. 3B. *CAV1, SLC7A11*, and *SLC7A5* were all negatively associated with miR-194.

## Relationships between oxidative stress genes and active UC pathogenesis

Based on the expression profiles of *CAV1, SLC7A11*, and *SLC7A5* in the GSE75214 and GSE48959 datasets, the R package ConsensusClusterPlus was used for cluster analysis. As illustrated in Figs. 4A and 4B & Table S5A, the GSE75214 dataset was clustered into two significantly different types. Active UC was mostly included in cluster 1, while inactive UC was included in cluster 2 (Fig. 4C, $P < 0.0001$). Similarly, the GSE48959 dataset was also clustered into significantly different two types (Figs. 4D and 4E); cluster 1 was markedly enriched with active UC, while cluster 2 was enriched with inactive UC (Fig. 4F, $P < 0.05$). These results indicated that *CAV1, SLC7A11*, and *SLC7A5* were strongly correlated with UC activity. Considering that active UC and inactive UC were more significantly enriched in the GSE75214 dataset than in the GSE48959 dataset, we therefore analyzed the roles of these three oxidative stress genes in different UC subtypes based on the GSE75214 dataset. We selected representative genes in clusters 1 and 2 that significantly interact with oxidative stress genes for GO analysis and KEGG enrichment analysis. We found that the function of cluster 1 was mostly related to inflammatory immune responses and had a

Peer

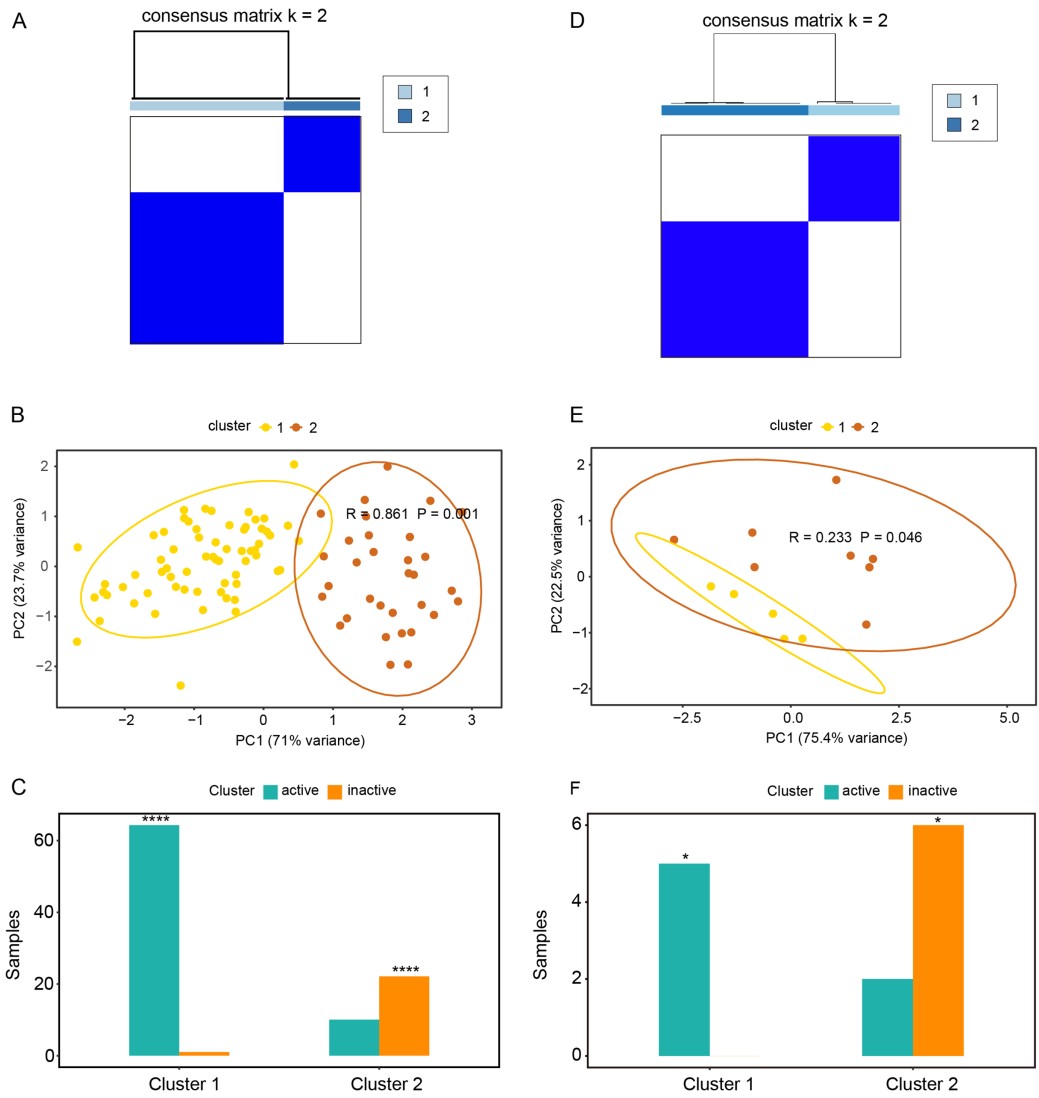

**Figure 4 Relationships between oxidative stress genes and active UC pathogenesis.** (A) Consensus clustering matrix of GSE75214 dataset for k = 2. Principal components analysis (B) and column chart (C) of active and inactive UC in the GSE75214 dataset. (D) Consensus clustering matrix of GSE48959 dataset for k = 2. Principal components analysis (E) and column chart (F) of active and inactive UC in the GSE48959 dataset. *$P < 0.05$, ****$P < 0.0001$.

relatively wide range of pathways (Table S5B). These were consistent with the finding that active UC was enriched in cluster 1. On the other hand, we found that cluster 2 was associated with a less inflammatory response (Table S5C); this was consistent with the enrichment of inactive UC in cluster 2. We further demonstrated that there was a significant enrichment of steroid hormone synthesis pathways in the functional pathway of cluster 2 (Table S5C). All these results suggest that the roles of oxidative stress genes in active UC may be related to immune inflammation, and that steroids may be effective for the treatment of active UC.

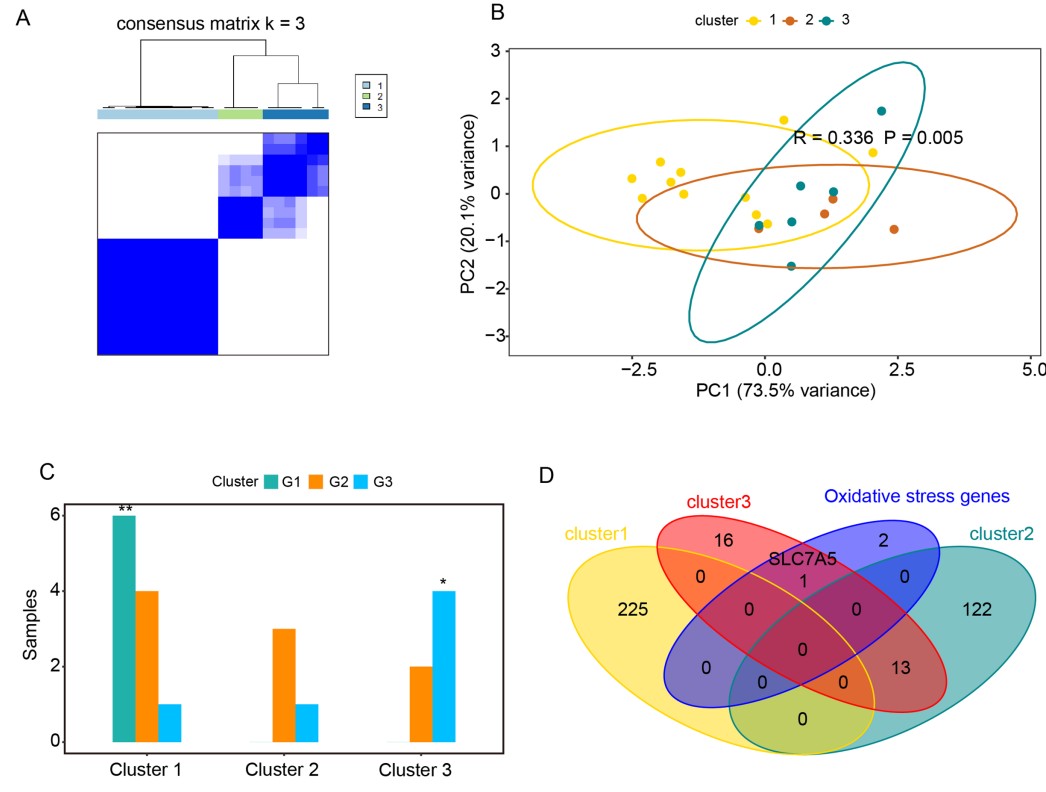

**Figure 5  Relationships between oxidative stress genes and glucocorticoid resistance in active UC.** (A) Consensus clustering matrix of GSE114603 dataset for k = 3. Principal components analysis (B) and column chart (C) of three groups of UC in the GSE114603 dataset. (D) Venn diagram for oxidative stress genes and representative genes of the three clusters. *$P < 0.05$, **$P < 0.01$.

## Relationships between oxidative stress genes and glucocorticoid resistance in active UC

The GSE114603 dataset containing glucocorticoid resistance genes was used to verify the relationships between oxidative stress genes and glucocorticoid resistance in active UC. Based on the expression profiles of *CAV1, SLC7A11*, and *SLC7A5* in the GSE114603 dataset, we clustered the GSE114603 dataset of patients with active UC into control (G1) and different response (G2, responders_day0; G3, non-responders_day0) groups using the R package ConsensusClusterPlus. As shown in Figs. 5A and 5B & Table S6A–S6C, we found that the GSE75214 dataset was clustered into three significantly different clusters. We also found that the control data were remarkably clustered into cluster 1, while non-responders_day0 data were clustered into cluster 3 (Fig. 5C, $P < 0.05$). The results indicated that *CAV1, SLC7A11*, and *SLC7A5* were associated with glucocorticoid therapy in patients with active UC. GO analysis and KEGG enrichment analysis were performed on representative genes that significantly interact with oxidative stress genes in cluster 1 (Table S6D), cluster 2 (Table S6E), and cluster 3 (Table S6F). The representative genes associated with oxidative stress interactions in cluster 1 were mostly related to cation reactions (such as zinc ion and copper ion), while in cluster 2, they were mostly related to

**Table 2 Drugs interaction with oxidative stress genes.**

| Search_term | Match_type | Drug | Interaction_types | Sources | Pmids |
|---|---|---|---|---|---|
| CAV1 | Definite | ALCOHOL | | NCI | 15845868 |
| CAV1 | Definite | TESTOSTERONE | | NCI | 11389065 |
| SLC7A11 | Definite | RILUZOLE | Inducer | TdgClinicalTrial | 10899284\|20226190\|12629173 |
| SLC7A5 | Definite | MELPHALAN | | PharmGKB | |

angiogenesis and wound healing. In cluster 3, there were related to pathways such as wound healing and in acute phase reactions. The most significant pathway in cluster 2 was extracellular matrix organization, while the most significant pathway in cluster 3 was extracellular matrix disassembly. Data from the non-responders_day0 group were mainly enriched in cluster 3; therefore, we speculate that the impact of oxidative stress genes on glucocorticoid therapy may be related to the stability of extracellular mechanisms.

### Prediction of drug interactions

Based on the relationships between the three oxidative stress genes and glucocorticoid therapy in active UC, we further explored drugs that interact with oxidative stress genes using the DGIdb database (https://www.dgidb.org/). We found that CAV1 interacted with alcohol, and testosterone, SLC7A11 interacted with riluzole, and SLC7A5 interacted with melphalan (Table 2). Among these three oxidative stress genes, *SLC7A5* was found to be a representative gene of cluster 3 (Fig. 5D). As cluster 3 significantly associated with glucocorticoid therapy resistance, we believed that finding drugs that interact with *SLC7A5* may be of great significance in improving resistance to glucocorticoid therapy active UC.

### Validation in mice with DSS-induced UC

HE staining was performed to monitor the effects of DXM on mice with DSS-induced UC. As illustrated in Fig. 6A, compared with the colonic samples of mice in the control group, a thickened muscle layer and an irregular mucosal layer were observed in the DSS and DSS + NS groups. In the treated group, these pathologic changes were alleviated to some extent. The expression of three oxidative stress-related genes (*CAV1, SLC7A11*, and *SLC7A5*) and miR-194 in colonic tissues was subsequently quantified. We found that the expression of *CAV1, SLC7A11*, and *SLC7A5* was significantly increased in mice with DSS-induced UC (Fig. 6B, $P < 0.05$) but was decreased following DXM treatment ($P < 0.05$). The opposite results were observed in the expression of miR-194 (Fig. 6C, $P < 0.05$). Furthermore, the levels of oxidative stress-related enzymes were determined. We demonstrated that compared with the control group, the levels of GSH and SOD were decreased in the DSS group (Figs. 6D and 6E, $P < 0.0001$), while both the GSSG and ROS levels were increased ($P < 0.05$). DXM treatment restored the decreased levels of GSH and SOD ($P < 0.0001$) but suppressed the increased levels of GSSG and ROS ($P < 0.01$). In addition, the mRNA expression of inflammatory cytokines was also determined. As shown in Fig. S3A, the IL-17, TNF-α, IL-1β, and IL-6 levels in the DSS model mice were dramatically upregulated relative to those in control mice ($P < 0.01$). DXM treatment considerably inhibited the

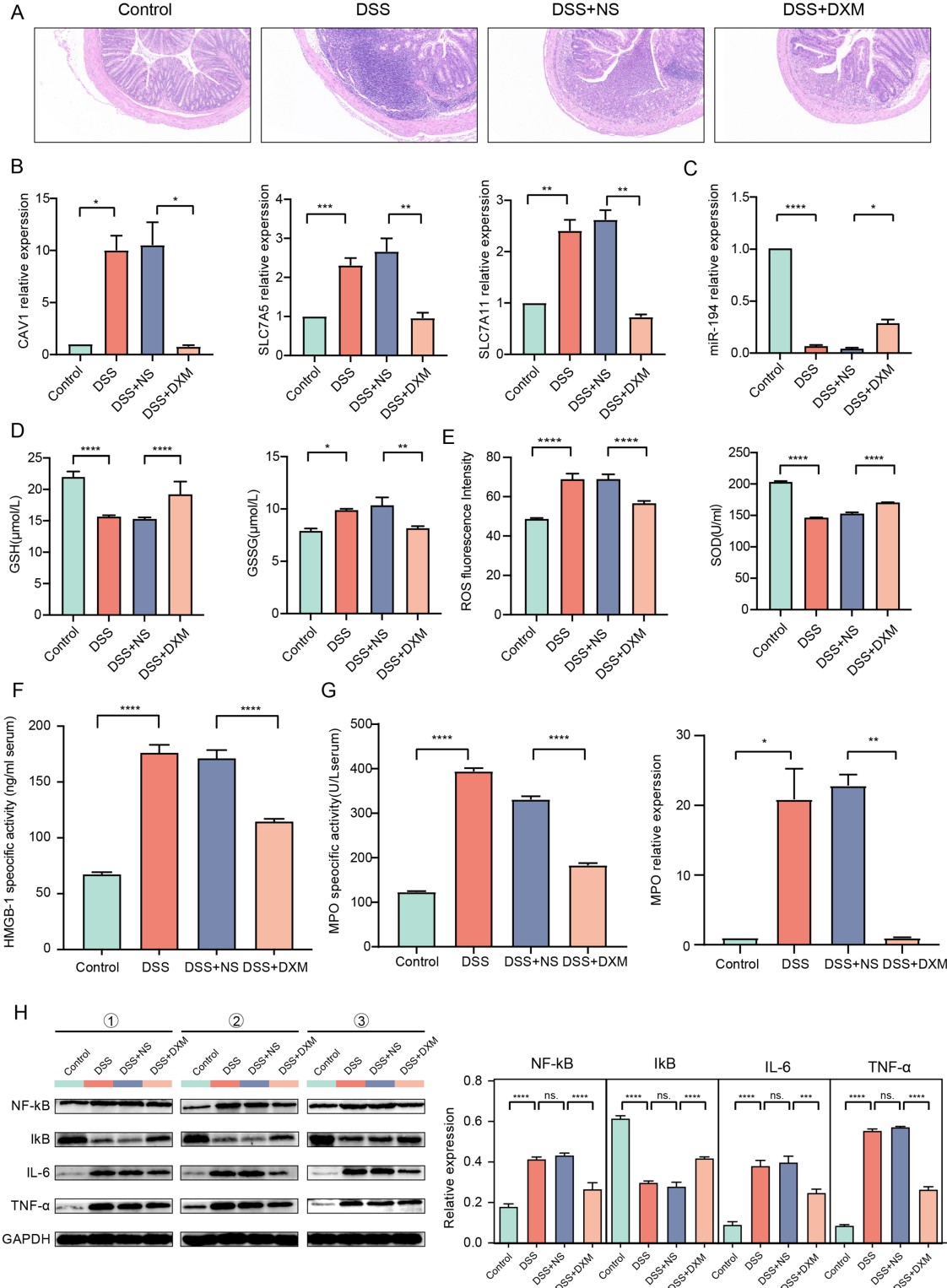

**Figure 6 Validation in DSS-induced UC mice.** (A) HE staining showed the histological changes in DSS-induced UC mice. The expression of *CAV1*, *SLC7A11*, *SLC7A5* (B) and miRNA-194 (C) in different groups. The levels of GSH, GSSG (D), ROS, SOD (E) and HMGB-1 (F) in different groups. (G) MPO activity or gene expression in different groups. (H) The protein levels of NF-κB, IκB, IL-6 and TNF-α in different groups. $^{*}P < 0.05$, $^{**}P < 0.01$, $^{***}P < 0.001$, $^{****}P < 0.0001$.

secretion of IL-17, TNF-α, IL-1β, and IL-6 ($P < 0.05$). The levels of pro-inflammatory effector factors (COX-2 and TLR4) were also upregulated in the DSS group compared with the control group (Fig. S3B, $P < 0.01$); however, compared with the DSS+NS group, the COX-2 and TLR4 levels in the DSS + DXM group were significantly downregulated ($P < 0.05$). As expected, the opposite patterns were observed in the expression of anti-inflammatory effector factors (Foxp3 and SOCS3). The expression products of HMGB-1 have pro-inflammatory activity, which can promote the infiltration and activation of various inflammatory cells into the intestinal mucosa and subsequently activate an inflammatory response in the intestinal mucosal tissues (Palone et al., 2016). High levels of HMGB-1 were found in the DSS group compared with those in the control group (Fig. 6F, $P < 0.0001$); however, compared with the DSS + NS group, the HMGB-1 level was decreased in the DSS + DXM group ($P < 0.0001$). MPO, a pro-inflammatory marker released from activated neutrophils, can caused damage at the site of inflammation in UC progression (Iwao et al., 2018). As illustrated in Fig. 6G, pronounced upregulation of MPO activity or gene expression was observed in DSS model mice compared with the control mice ($P < 0.05$). However, DXM treatment significantly decreased the activity and expression of MPO in mice with UC ($P < 0.01$). NF-κB is an essential transcription factor for the transcription and translation of a series of pro-inflammatory genes, including IL-6 and TNF-α, while IκB is a key suppressor of NF-κB activation (Xie et al., 2020). As shown in Fig. 6H, high protein levels of NF-κb, IL-6, and TNF-α were observed in DSS model mice compared with control mice ($P < 0.0001$), and DXM treatment significantly reduced these protein levels ($P < 0.01$). As expected, the opposite results were observed in the protein levels of IκB ($P < 0.01$).

## DISCUSSION

UC is a chronic inflammatory disease, and imbalanced regulation of oxidative stress plays a crucial role in its pathology (Rana et al., 2014; Jena, Trivedi & Sandala, 2012). Clinical interventions are mainly focused on alleviating the symptoms of patients UC patients because of its complex pathogenesis and mechanism (Ungaro et al., 2017; Sarvestani et al., 2021). Side effects and drug resistance are still inevitable (Nakase et al., 2021; Magro et al., 2017; Truelove & Witts, 1955). The lncRNA-miRNA-mRNA regulatory network has been suggested to play a vital role in regulation of oxidative stress in the pathogenesis of UC and glucocorticoid-related disorders. A previous study conducted by Wang et al. showed that lncRNA MEG3 acts as a ceRNA for IL-10 via sponging miR-98-5p to relieve inflammation and oxidative stress in a UC rat model (Wang et al., 2021). DXM is a well-known glucocorticoid drug that is widely used in the treatment of UC; however, it leads to severe osteoporosis. Liu et al. (2018) found that FGF1 could enhance the efficiency of DXM through the lncRNA GAS5/miR-21 axis. Therefore, the construction of ceRNA and further exploration of oxidative stress-related genes underlying the pathogenesis and glucocorticoid resistance mechanism of UC are essential to understand potential biomarkers in UC.

In the current study, DE-lncRNAs, DE-miRNAs, and DE-mRNAs were identified in UC and control tissues from the GSE75214 and GSE48959 datasets. Based on the miRDB,

miRTarBase, and TargetScan databases, each DE-miRNA's predicted mRNA was obtained. In addition, we used R cor.test to calculate the correlations between DE-lncRNAs and DE-mRNAs and between DE-miRNAs and DE-mRNAs and then constructed a DE-lncRNA-DE-mRNA network and a DE-miRNA-DE-mRNA network. After that, according to the intersection-DE-mRNAs, a DE-lncRNA-DE-mRNA network and a DE-miRNAs-DE-mRNA network were combined to construct a DE-lncRNA-DE-miRNA-DE-mRNA network including 30 lncRNAs, 3 miRNAs, and 19 mRNAs. The GSE75214 and GSE48959 databases were validated by AUC analysis to determine whether these 19 mRNAs from ceRNA were important in the pathogenesis of UC. We then further screened three oxidative stress-related mRNAs that can interact with miR-194, namely, *CAV1*, *SLC7A11*, and *SLC7A5*. *SLC7A5* was considered a representative gene associated with glucocorticoid therapy resistance.

The *SLC7A5* gene, which is mapped at 16q24.2, has 39,477 nucleotides with 10 exons (*Scalise et al., 2018*). SLC7A5 protein, which is also known as L-type amino acid transporter 1 (LAT1), is an amino acid transporter with 12 transmembrane α-helices and has been confirmed to regulate the distribution of specific amino acids across cell membranes (*Scalise et al., 2018*; *Napolitano et al., 2017*). In general, SLC7A5 is highly expressed in the inner blood retinal barrier, blood-brain barrier, and brain endothelial cells (*Boado et al., 1999*; *Tomi et al., 2005*). Interestingly, in several human cancers, SLC7A5 has also been observed to be overexpressed (*Zhao, Wang & Pan, 2015*). Therefore, SLC7A5 has been proposed as a novel target for the treatment of human cancers. An obvious example is that JPH203/KYT-0353, an inhibitor of SLC7A5, can clinically suppress tumor growth (*Oda et al., 2010*). SLC7A5 also plays a crucial role in the immune microenvironment. For example, in macrophages, SLC7A5 can mediate the transport of leucine to promote the secretion of proinflammatory cytokines *via* mTORC1 signaling (*Yoon et al., 2018*). In the progression of rheumatoid arthritis, pronounced upregulation of SLC7A5 has been observed in monocytes and is negatively correlated with the prognosis (*Yoon et al., 2018*). SLC7A5 is also essential for maintaining the functions of NK cells (*Loftus et al., 2018*). Oxidative stress is involved in various human disorders and is harmful to human health. As a potent neurotoxin, methylmercury (MeHg) can induce oxidative stress and cell apoptosis. *Granitzer et al. (2021)* reported that knocking-down *SLC7A5* enhances the oxidative stress caused by MeHg in HTR-8/SVneo cells. *Brahmajothi et al. (2014)* reported that hyperoxia and increased oxidative stress have adverse effects on the expression and function of *SLC7A5* in alveolar epithelial cells. In the current study, we identified *SLC7A5* as a representative oxidative stress-related genes in UC and further confirmed that it was upregulated in a UC mouse model. This is in accordance with the findings of a previous report conducted by *Al-Mustanjid et al. (2020)* who found that SLC7A5 is a transcription factor that is upregulated in inflammatory bowel disease. We also found that DXM treatment significantly decreased *SLC7A5* expression, which further validated the potential role of *SLC7A5* in glucocorticoid therapy. In addition, based on GO analysis and KEGG enrichment analysis, we found that SLC7A5 was closely associated with extracellular matrix disassembly. Similarly, *Yoon et al. (2018)* demonstrated that SLC7A5 can mediate leucine influx from extracellular matrix into cells, leading to a decreased rate of

extracellular acidification. These results imply that *SLC7A5* may be related to the stability of extracellular mechanisms. Furthermore, the prediction of drug interactions indicated that SLC7A5 can effectively interact with melphalan. It is well-known that melphalan is an effective clinical drug for breast cancer (BC) treatment. Interestingly, the protein levels of SLC7A5 are higher in estrogen receptor (ER)-positive BC cells than in ER-negative BC cells. These results imply that SLC7A5 is not only an underlying target for BC treatment but also a potential biomarker for UC. In addition, we suggest that melphalan may also have the potential to improve the resistance of active UC to glucocorticoid therapy.

Some validations were conducted in mice with DSS-induced UC. A thickened muscle layer and an irregular mucosal layer were observed in the colonic tissues of mice with DSS-induced UC, suggesting that the UC model was established successfully. As expected, the pathologic changes were noted to be in remission following DXM treatment, indicating that DXM is effective for the treatment of UC. Apart from *SLC7A5*, the other two oxidative stress-related genes, *CAV1* and *SLC7A11*, were found to be upregulated in the UC model mice and downregulated after DXM treatment. The opposite results were observed in terms of miR-194 expression. The abovementioned data validated the results of the bioinformatics analysis. Moreover, the contents of oxidants (GSSG and ROS) and antioxidants (GSH and SOD) in UC model mice were also determined and suggested the antioxidative effect of DMX in UC progression. Inflammatory responses are considered as one of the main factors that affect UC progression. Some inflammatory cytokines are reported to be involved in the pathogenesis of UC. For example, IL-17 can induce inflammation by recruiting leukocytes (*Iwakura et al., 2011*). By affecting the secretory function of intestinal epithelial cells, IL-6 aggravates the progression of UC (*Nishida et al., 2018*). The immunoreactive TNF-α protein is also closely associated with the development of active UC (*Popivanova et al., 2008*). Furthermore, upregulation of pro-inflammatory effector factors such as TLR4 can produce inflammatory cytokines including TNF-α, IL-1β, and IL-6 (*Rosales-Martinez et al., 2016*; *Ye et al., 2017*), while anti-inflammatory effector factors have inhibitory effects on inflammatory cytokine secretion. For instance, Foxp3, a nuclear transcription factor of Treg cells, plays a key role in releasing anti-inflammatory cytokines (*Tang et al., 2019*; *Bin Dhuban et al., 2019*). SOCS3 belongs to a family of intracellular proteins that negatively regulate inflammation (*Yoshimura, Naka & Kubo, 2007*). In this study, the increased levels of inflammatory cytokines (IL-17, TNF-α, IL-1β, and IL-6) and pro-inflammatory effector factors (COX-2 and TLR4) caused by DSS stimulation were significantly reduced following DXM treatment. DXM treatment was also confirmed to increase the expression of Foxp3 and SOCS3. These results imply that DXM may alleviate the development of UC *via* inhibiting inflammation. In the development of inflammatory diseases, mature neutrophils can release MPO to interact with macrophages, initiating a series of molecular cascades and the secretion of inflammatory cytokines (*Lefkowitz & Lefkowitz, 2001*). As expected, high levels of MPO activity or gene expression were observed in mice with DSS-induced UC, which was similar to the findings of a previous study that showed that MPO activity was dramatically upregulated in rats with acetic acid-induced colitis (*Ghatule et al., 2012*). We also found that DXM showed a suppressive effect on MPO activity. NF-κB serves as a crucial

transcription factor responsible for the activation of various pro-inflammatory genes, such as IL-6 and TNF-α, whereas IκB acts as a pivotal suppressor in regulating NF-κB activation (*Xie et al., 2020*). We further demonstrated that DXM could inhibit the protein levels of NF-κB, IL-6, and TNF-α but was associated with increased IκB levels. These results further validated the anti-inflammatory role of DXM in UC progression.

Some limitations of this study should be acknowledged. First, more clinical samples from patients with UC should be collected to validate the oxidative stress genes and measure gut microbiota data. Second, other than melphalan, we found that CAV1 interacted with testosterone and that SLC7A11 interacted with riluzole. The clinical applications of these drugs and drug targets are needed. Third, whether these oxidative stress genes play driving roles or just act as bystanders in the pathogenesis of UC may be an interesting subject in the future.

## CONCLUSION

In conclusion, a lncRNA-miRNA-mRNA network containing 30 DE-lncRNAs, 3 DE-miRNAs, and 19 DE-mRNAs was established by bioinformatics analysis. Among the 19 DE-mRNAs, 3 mRNAs including *CAV1, SLC7A11*, and *SLC7A5* were identified as oxidative stress-related genes, and *SLC7A5* was considered a representative gene associated with glucocorticoid therapy resistance, which was further validated in animal experiments. These results highlight the effects of oxidative stress-related genes on the pathogenesis and glucocorticoid therapy mechanism of UC and may provide a new therapeutic target for the treatment of UC in the clinic.

### Funding
This work was supported by National Natural Science Foundation of China (Project No: 81603509), Natural Science Foundation of Shandong Province (Project No: ZR2014HQ051), Key Research and Development Plan of Jining (Project No: 2021YXNS007). The funders had no role in study design, data collection and analysis, decision to publish, or preparation of the manuscript.

### Grant Disclosures
The following grant information was disclosed by the authors:
National Natural Science Foundation of China: 81603509.
Natural Science Foundation of Shandong Province: ZR2014HQ051.
Key Research and Development Plan of Jining: 2021YXNS007.

### Competing Interests
The authors declare that they have no competing interests.

### Author Contributions
- Qifang Li analyzed the data, prepared figures and/or tables, authored or reviewed drafts of the article, and approved the final draft.

- Yuan Liu performed the experiments, analyzed the data, authored or reviewed drafts of the article, and approved the final draft.
- Bingbing Li analyzed the data, authored or reviewed drafts of the article, and approved the final draft.
- Canlei Zheng performed the experiments, prepared figures and/or tables, authored or reviewed drafts of the article, and approved the final draft.
- Bin Yu conceived and designed the experiments, performed the experiments, analyzed the data, prepared figures and/or tables, authored or reviewed drafts of the article, and approved the final draft.
- Kai Niu analyzed the data, prepared figures and/or tables, authored or reviewed drafts of the article, and approved the final draft.
- Yi Qiao analyzed the data, prepared figures and/or tables, authored or reviewed drafts of the article, and approved the final draft.

### Animal Ethics

The following information was supplied relating to ethical approvals (*i.e.*, approving body and any reference numbers):

Jining Medical University provided full approval for this research (approval number: JNMC-2023-DW-090).

### Data Availability

The data is available at NCBI GEO: GSE75214, GSE48959, GSE114603.

The raw data are available in the Supplemental Files and at figshare: Yu, Bin (2023). Supplementary Table S1-6.xlsx. figshare. Dataset. https://doi.org/10.6084/m9.figshare.24188217.v3.

### Supplemental Information

Supplemental information for this article can be found online at http://dx.doi.org/10.7717/peerj.17213#supplemental-information.

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
