# Peer review of "Bioinformatics analysis of oxidative stress genes in the pathogenesis of ulcerative colitis based on a competing endogenous RNA regulatory network"

_PeerJ, doi:10.7717/peerj.17213_

## Round 0.1 · original submission · Major Revisions

Please address the concerns of all reviewers and amend the manuscript accordingly.

**Language Note:** The review process has identified that the English language must be improved. PeerJ can provide language editing services - please contact us at [email protected] for pricing (be sure to provide your manuscript number and title). Alternatively, you should make your own arrangements to improve the language quality and provide details in your response letter. – PeerJ Staff

·

Basic reporting

The authors should have measured the oxidative stress genes and the competing endogenous RNA regulatory network in animal models of UC and in patients with UC and provided that data to substantiate their present work.

Measurement of oxidative stress genes in animal models of UC and in patients with UC such as ROS, NF-kB, IkB, IL-6, TNF, HMGB1, glutathione and other antioxidants is needed. GUt microbiota data and the measurement of oxidative stress and regulatory RNA network in gut of the animal models of UC and if possible in patients is needed.

In vitro and bioinformatics data and collection of data form data bases is interesting but provide very little practical information that is applicable to patients and much less to know the drug targets.

Experimental design

Ok but measurement of the same indices in animal models of UC and patients samples is needed.

Validity of the findings

Findings need to be validated in animal models of UC and patients.

Reviewer 2 ·

Basic reporting

● Writing:
The frequent usage of acronyms in the paper might confuse readers. I strongly recommend reducing the use of acronyms, unless when absolutely necessary.

The author has a habit of writing long sentences packing multiple points, which makes the sentences difficult to read. I suggest the author try simplifying the sentences by breaking them into short ones, each of which conveys one clear point.

Some sentences are written in rather awkward English. I suggest consulting fluent speakers to help revise the language, mostly the wording, for the revision.
● Figures:
Fonts across figures are not consistent, and most of the fonts are too small to be legible.

The labelings of the result panels, such as axis labels and figure legends, are overly simplified; the author should consider making figures more self-explanatory.

Experimental design

● Major comments:
What are the relative contributions of oxidative stress-related genes vs other identified differentially expressed genes? Does the author envision oxidative stress genes playing a driving role or just as bystanders?

What is the relationship between the set of identified oxidative stress-related genes with other identified differentially expressed genes? Do they exhibit any mutual regulation?
● Figure-by-figure comments:
● Fig. 1
The author didn’t explain the differences between the two selected datasets. It would be nice to include explanations for the rationale for choosing these particular datasets in the main text.
● Fig. 2
What is the concordance between the two datasets in terms of the DEGs? It appears from the figure that the differentially expressed mRNAs from the two datasets (GSE75214 and GSE48959) significantly differ. Should one expect the two datasets to be similar to each other?
● Fig. 3
What is the logical connection between this figure and the rest of the study? It appears that oxidative stress-related genes are the main focus of this study, and therefore this figure seems to be a digression from the main point, or at best just a demonstration of the ceRNA network identification algorithm, which is developed by published studies. As a result, I suggest the author either eliminate this figure or state clearly in the text the main connection of this figure to the main point of this study.
● Fig. 5 & 6
Are oxidative-stress-related genes the best classifiers for different types of UCs? How does the clustering using oxidative stress-related genes compare with clustering using other set of differentially expressed genes identified in Fig. 3?
Similarly, since the author has built a random forest classifier with 19 differentially expressed genes identified in Fig. 3 (line 249 of main text), would it be possible to build a similar classifier, but using only the oxidative stress-related genes and compare the quality of the two classifier?
● Fig. 7
Since the author mentioned GR-resistance as a major challenge in UC treatment, is it possible to conduct the same expression test on tissues samples from mouse with DXM-resistance (if such model exist)?

What statistical test were used for the qPCR results?

Validity of the findings

● In the main Result section at the beginning of the paper (line 36-46 of main text), the author made one claim “Meanwhile, the impact of oxidative stress on glucocorticoid therapy may be related to the stability of extracellular mechanisms.” This clain remained largely speculative, as it is not supported by any experimental evidence. I suggest authors either eliminate such a claim or add more experimental data supporting this claim.
● Is it possible to provide any experimental evidence for the extracellular matrix disassembly hypothesis?

Reviewer 3 ·

Basic reporting

It is comprehensive study. Authors narrowed down target genes using bioinformatics tools and crosschecked with different programs. The manuscript elucidates possible lncRNA-miRNA-mRNA network and suggest underlying mechanisms of UC supported with animal study.

Experimental design

1. Why is the spearman correlation used for ceRNA network construction not pearson correlation?
2. Supplementary fig. 1, what are the eigenvalues of two main components?
3. Figure 7, as to miRNA expression in tissue samples, were the tissue samples collected from lesions?

Validity of the findings

1. Did authors check protein expression of tested pro- and inflammatory markers by Western Blot?

Additional comments

1. At line 25, what does it mean by ‘network’? Network of lncRNA-miRNA-mRNA in terms of ceRNA regulatory network in UC? Or in general? It is not as clear as line from 109 to 111.
2. Grammar errors and typo
a. For example, ‘A, B, and C’ means different from ‘A, B and C’ (line 68, 77, and etc)
b. Fix LncRNA to lncRNA
3. Fix line 152. Having space between plus and ‘(‘ on line 153 would fix it

---

## Round 0.2 · accepted · Accept

All issues pointed out by the reviewers were adequately addressed and the revised manuscript is acceptable now.

·

Basic reporting

In the present study the authors showed that RNA regulatory network is involved in steroid resistance/effectiveness in UC.
Authors have answered all the criticisms adequately. in the revised version.

Experimental design

ok

Validity of the findings

ok

Additional comments

nil

Reviewer 2 ·

Basic reporting

All comments have been properly addressed.

Experimental design

All comments have been properly addressed.

Validity of the findings

All comments have been properly addressed.

Additional comments

All comments have been properly addressed.

Reviewer 3 ·

Basic reporting

Authors replied to reviewer's comments well

Experimental design

Comparison of animal model to UC patients are not complete. Authors need to work on justify lack of UC patient's index

Validity of the findings

Authors replied to reviewer's comments well